# Rapid detection of brucellosis using a quantum dot-based immunochromatographic test strip

Guangqiang Li[1,2], Zhen Rong[3], Shengqi Wang[3], Hongyan Zhao[1], Dongri Piao[1], Xiaowen Yang[1], Guozhong Tian[1], Hai Jiang[1]*

1 State Key Laboratory for Infectious Disease Prevention and Control, Collaborative Innovation Center for Diagnosis and Treatment of Infectious Diseases, National Institute for Communicable Disease Control and Prevention, Chinese Center for Disease Control and Prevention, Beijing, China, 2 Center for Disease Control and Prevention, Western Theater Command, Lanzhou, China, 3 Beijing Institute of Radiation Medicine, Beijing Key Laboratory of New Molecular Diagnosis Technologies for Infectious Diseases, Beijing, China

* jianghai@icdc.cn

**Data Availability Statement:** All relevant data are within the manuscript and its Supporting Information files.

## Abstract

Novel diagnostic tools are a major challenge for brucellosis research, especially in developing countries. Herein, we established a handheld quantum dot (QD) immunochromatographic device for the fast detection of brucellosis antibodies in the field. Total bacterial protein extracted from *Brucella* 104M served as labelling and coating antigen. QD labelling and immunochromatography methods were used to optimise reaction conditions, labelling conditions, reaction temperature and storage temperature. QD test strips were employed to test brucellosis serum to determine their sensitivity, specificity and stability. Test strips were compared with Rose Bengal test, standard agglutination test and colloidal gold immunochromatographic assay. Labelled *Brucella* total protein displayed good specificity and no cross-reactivity. The concentration of labelled total bacterial protein was 3.9 mg/ml, the coating concentration was 2.0 mg/ ml, and the serum titre with the lowest detection sensitivity was 1:25. The optimal reaction temperature for the test strip was 25−30˚C. The test strip was stable after storage at room temperature and the repeatability was high, with a coefficient of variation of 4.0%. After testing 199 serum samples, the sensitivity of the QD test strip was 98.53%, the specificity was 93.57%, and the coincidence rate with the standard agglutination test was 96.98%. The developed QD immunochromatographic method can be used for rapid detection and preliminary screening of brucellosis in the field.

## Author summary

Brucellosis is a neglected infection that has a widespread geographic distribution. Based on an evaluation from the World Health Organization (WHO), brucellosis cases have been reported in more than 170 countries with about 500,000 new cases being reported each year. However, the actual number of brucellosis patients is much higher, and it is believed to be approximately 10–25 times the number of reported cases. This big discrepancy between the reported rate and the actual incidence rate is largely due to misdiagnosis

**Funding:** This study was funded by a Major Infectious Diseases such as AIDS and Viral Hepatitis Prevention and Control Technology Major Projects grant (grant no. 2018ZX10712-001). The funders had no role in study design, data collection and analysis, decision to publish, or preparation of the manuscript.

**Competing interests:** The authors have declared that no competing interests exist.

and underdiagnosis, especially in endemic areas. In recent years, considerable effort has been mobilized toward the development of rapid, reliable field diagnostic assays and molecular diagnostic approaches. In this study, we established a handheld quantum dot (QD) immunochromatographic device for the fast detection of brucellosis antibodies in the field. This QD test strip does not require extensive laboratory infrastructure or technical expertise, and has high sensitivity (98.53%) and specificity (93.57%), as well as a high coincidence rate (96.98%) with SAT. This new tool is expected to provide fast and simple point-of-care testing at county-level clinics and CDC labs.

## Introduction

Brucellosis, an infectious disease caused by bacteria of the genus *Brucella*, is one of the most widespread zoonotic diseases [1]. This disease causes great harm to humans and animals, and it has been reported in more than 170 countries, with particular prevalence in the Mediterranean, the Americas, Asia and Africa [2]. The World Organisation for Animal Health (OIE) has classified brucellosis as a Class B infectious disease, and this disease results in ~500,000 new cases each year worldwide [3]. In China, *B. melitensis* has become the predominant causative strains for brucellosis and shows strong virulence. Humans are mainly infected by contact with infected animals or the consumption of unsterilised dairy products. *Brucella* has been also used as biological warfare agents [4].

Brucellosis is treatable, but early diagnosis followed by timely medical intervention is the key. Currently, the diagnosis of brucellosis in China relies heavily on serum detection tests[5], which include Rose Bengal test (RBT), colloidal gold immunochromatographic assay (GICA), standard agglutination test (SAT), complement fixation test (CFT) and enzyme-linked immunosorbent assay (ELISA). In addition, the quantum dot (QD) immunochromatographic method has great potential: this technology is simple to operate and has been shown to have high sensitivity and specificity for the rapid detection of brucellosis in the field or clinical setting.

A QD is an important semiconductor nanocrystalline material, which is generally spherical or quasi-spherical in shape, with a diameter between 2 and 10 nm. It is a new type of fluorescent labelling material with a large Stokes shift, good chemical stability, a wide excitation wavelength range, a narrow emission spectrum, strong light stability, and a long fluorescence lifetime. Typically, QD is composed of materials such as Group IV elements, Group II–VI elements (such as CdSe, CdTe, ZnS and ZnSe) and Group III–VI elements (such as InP and InAs). The above materials constitute the core/shell structures of QD, examples of which include CdSe/ZnS and CdSe/ CdS [6]. In this study, we chose the CdSe quantum dots with a core shell structure, as these have good luminous performance and strong stability. Due to its unique optical and physical properties, QD is attracting increasing attention from scientific researchers, and has great potential for applications in biology and medicine to improve detection sensitivity and stability [7,8,9]. In particular, QD probe preparation for deployment in biological and medical immunity applications has been undergoing rapid development, especially for the detection of *Vibrio parahaemolyticus* [10] and C-reactive protein, and rapid diagnosis of infectious diseases such as African swine fever and Middle East respiratory syndrome [11–12].

In the present study, we engineered an immunochromatographic test strip combining QD labelling and immunochromatographic technologies. When the double-antigen sandwich structure with a nitrocellulose membrane as carrier binds to *Brucella* total protein, the QD is

excited and elicits a color response. The assay developed here can be employed for the rapid initial screening of *Brucella* antibodies in the serum of suspected patients and animals.

## Materials and methods

### Ethics statement and study samples

This study involves a retrospective analysis of brucellosis patients whose serum samples were collected at the Tongliao Center for Endemic Disease between August 2018 and November 2019. Brucellosis patients with a clear clinical history and experimental diagnosis were included. The diagnostic criteria were in strict accordance with the published "Diagnostic Criteria for Brucellosis (WS 269–2019)". The study protocol was reviewed and approved by the China CDC according to their guidelines for studies on human subjects. No identifying information about the patients will be collected or reported as part of this project.

### Materials

The nitrocellulose (NC) membrane (cat. No. HF13502S25), glass cellulose membrane (cat. No. Ahlstrom8964), serum sample pad (cat. No. XQ-Y2), absorbent paper (cat. No. H5072) and PVC bottom plate (cat. No. DB-6) were purchased from Shanghai Jieyi Biotechnology Co., Ltd. The machine we used for spraying the C and T lines is a XY1000 dot film meter produced by Bio- Dot in the United States. Bovine serum albumin (BSA), staphylococcal protein A (SPA), phosphate-buffered saline (PBS), PBS containing Tween-20, mercaptosuccinic acid (MES), N-hydroxysuccinimide (NHS), 1-ethyl-(3-dimethylaminopropyl) carbodiimide hydrochloride (EDC) and fetal bovine serum were purchased from Sigma-Aldrich. QDs (surface active group = carboxyl; cat. No. FM610C; NanoGen). phosphate buffer, Tween-20, sucrose, absolute ethanol, sodium chloride and methanol were purchased from Sinopharm Chemical Reagent Co., Ltd. RBT and SAT antigens, colloidal gold immunochromatographic test strips, sheep brucellosis standard serum (serum titer, 1:200), and healthy human serum were provided by the National Institute for Communicable Disease Control and Prevention. *Vibrio cholerae* antigen was provided by the Institute of Radiation Medicine, Military Medical Research Institute of the Academy of Military Sciences. Water (18.2 mΩ) was from a Milli-Q pure water system.

### Methods

#### Preparation of *Brucella* total protein

*Brucella* 104M with good antigenicity was selected to prepare total protein. The culture grown on medium was placed in a water bath at 70–80˚C for 1 h. After centrifugation, the supernatant was discarded and the cell pellet was resuspended in 0.5% carbamic acid physiological saline so that the concentration of the suspension was more than twice that of the *Brucella* SAT antigen stock. The sample was incubated under a steam pressure of 108˚C for 40–60 min, and then left for more than 2 weeks. The supernatant was centrifuged, precipitated, and extracted. The supernatant was sterile filtered, and the filtrate served as *Brucella* total protein.

#### Preparation of brucellosis QD immunochromatographic test strips

A 25-μl sample of QD 610 nm solution, 5 μl of 0.1 M MES (pH 6.0), and 20 μl of ultrapure water were added to a 1.5-mL centrifuge tube, mixed, and centrifuged for 10 s. EDC cross-linking activator EDC (-20˚C) and NHS in ultrapure water were used at concentrations of 1.9 mg/mL and 2.1 mg/mL, respectively. A 5-μl sample of EDC solution and 7.5 μl of NHS solution were mixed and incubated in a 37˚C incubator for 15 min, and ultrasonic dispersion was

performed. After centrifuging at 8000 g for 20 min, the supernatant was pipetted thoroughly and 25 μl of 0.01 M MES (pH 6.0) was added. Next, the QD marker was combined with the antigen, and after activation, blocking agent was added and incubated for 3 h. The original volume of resuspension solution was added, mixed well, and set aside. This solution was diluted 150-fold, freeze-dried at –60°C for 3–4 h, and stored in a sealed container at room temperature for future use.

SPA was diluted to 1.0 mg/mL and used for the C line, and *Brucella* total bacterial protein was diluted to 2.0 mg/mL and used for the T line. After optimising parameters, the NC film relative humidity was 20% and drying was performed at 37°C for 2–3 h. Test strips were assembled by overlapping the bottom plate, the NC membrane, the bonding pad, the sample pad, and the water absorption pad by 1 to 1.5 mm.

## Test strip detection method and principle

A 10-μl sample was diluted to 100 μl with the sample diluent, and 70 μl was placed on the test strip sample pad. After 10 min, the pad was observed under a handheld UV lamp and a dry immunofluorescence analyser. The results were measured at the T line.

When *Brucella* antibodies are present in the sample, QDs on the release pad bind to the antibodies to form QD-Ag-antibody immune complexes. When flowing past, the immune complex binds to the antigen attached at the detection line. When a specific immune response occurs, some of the QDs form a T line. Meanwhile, immune complexes that do not react with antigen continue to migrate forward, and are captured by the fixed secondary antibody (SPA) attached at the control line, and a specific immune response occurs, leaving excess QDs at the quality control line to form the C line. Any QDs will continue to migrate toward the absorbent paper, and eventually reach the absorbent pad and the test solution. The strip design is shown in Fig 1.

## Optimisation of test strip conditions

A 25-μl sample of QD served as the basic dosage, and the molar ratios of EDC:NHS:QD were 2000:3000:1, 2000:4000:1, 3000:2000:1, and 4000:2000:1. The four different molar ratios were analysed in terms of fluorescence intensity at the detection line after QDs were labelled with antigen. Three antigen coupling amounts (10, 15 and 20 μg) and two coupling durations (1 and 3 h) were tested. BSA was dissolved in glycine solution (1% BSA) as a blocking agent to optimise blocking conditions.

Several sheets of binding pads were employed for treatment 1 which consisted of 0.1% BSA and 0.5% Tween-20. Treatment 2 consisted of 5% sucrose and 0.6% Tween-20, and two sheets of binding pads. The results were compared for treated and untreated binding pads. QD antigen conjugates were diluted 100-fold, 150-fold, 200-fold, 300-fold, 400-fold and 500-fold, lyophilised, and assembled into test strips for detection.

Three dilutions were tested (methanol, 0.01 M PB in PBS, 10% isopropanol in PBS). Strips were dried at 37°C for 2–3 h, *Brucella* total protein (3.9 mg/mL) was diluted to 1.0 mg/mL, 2.0 mg/mL and 2.5 mg/mL, and the effects of the three coating concentrations on the T-ray signal were observed. SPA solution and *Brucella* total bacterial protein solution were placed into the wiper, the C-line of the nozzle and the wiper was adjusted to 1.0 μl/cm, and the T-line was set to 1.0 μl/cm, 0.6 μl/cm, and 0.5 μl/cm. After drying, when set to 0.6 μl/cm, there was no imprint on the T-line of the NC membrane, and diffusion was not observed.

Test strips were reacted at 4, 22, 25, 28, 31, 34 and 37°C, and the intensity of the T-line signal at each reaction temperature was measured. The specificity of the test strip was tested using normal human negative control serum, anthrax serum, cholera serum, hydatid patient

serum, and brucellosis serum. Positive sera were diluted to a titre of 1:3200, and this was diluted 2-, 4, 8-, 16-, 32-, 64-, 128- and 256-fold to investigate the signal detection limit of the test strip.

Five samples from the same batch of test strips were maintained at room temperature and randomly sampled every 5 days, and serum from positive brucellosis samples was added drop-wise to calculate the average, standard deviation and coefficient of variation, and the stability time and decay of the T-value signal over time.

### Method evaluation for screening

QD immunochromatography, GICA, RBT and SAT were compared to determine the authenticity and value of the four methods for the detection of brucellosis antibodies.

### Statistical analysis

SPSS 21.0 was used for statistical analysis, and statistical indicators such as the average and coefficient of variation were compared. Chi-square tests were performed on paired data for QD immunochromatographic test strips, GICA, SAT and RBT. Test data were processed according to the evaluation method of the screening test, and the screening effect was evaluated at the same time.

## Results

### Optimisation of immunochromatographic test strips

The C-line appeared on all five test strips, confirming that the results are valid. At the T line position, the specificity of negative serum was as expected. The fluorescence intensity of the T-line was brightest when the molar ratio of EDC:NHS:QD was 2000:3000:1, followed by 3000:2000:1, and weakest at 4000:2000:1 (Fig 2A).

### Optimisation of antigen labelling

When the C-line appeared, the negative control exhibited better specificity. The T-line after a 3 h coupling was brighter than after a 1 h coupling (Fig 2B). Using a QD volume of 25 μl, the

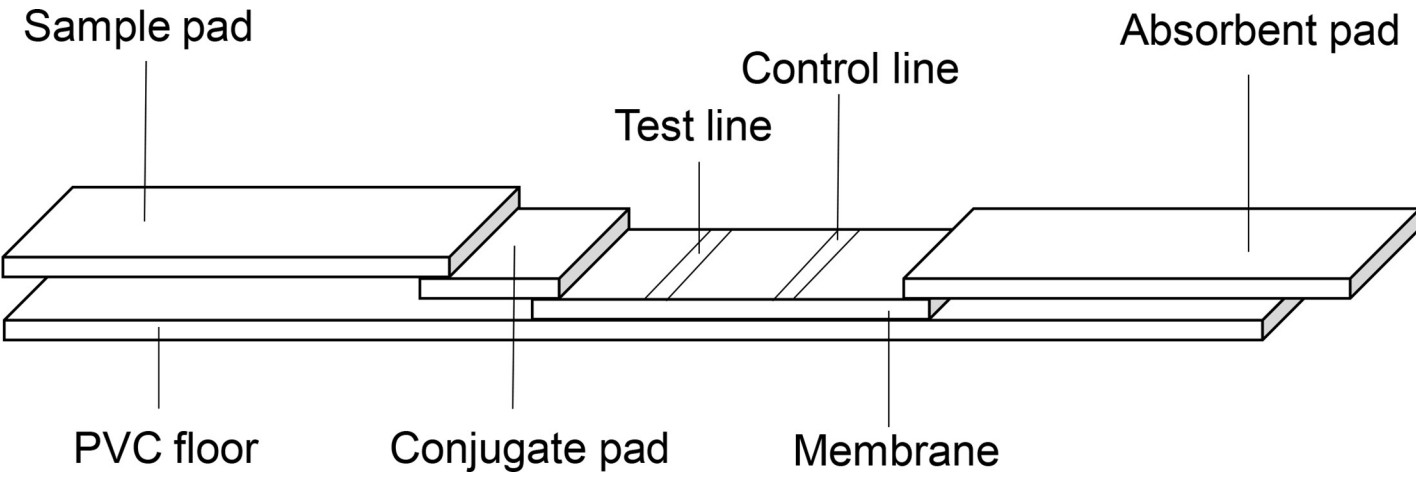

**Fig 1. Immunochromatographic test strip design.**

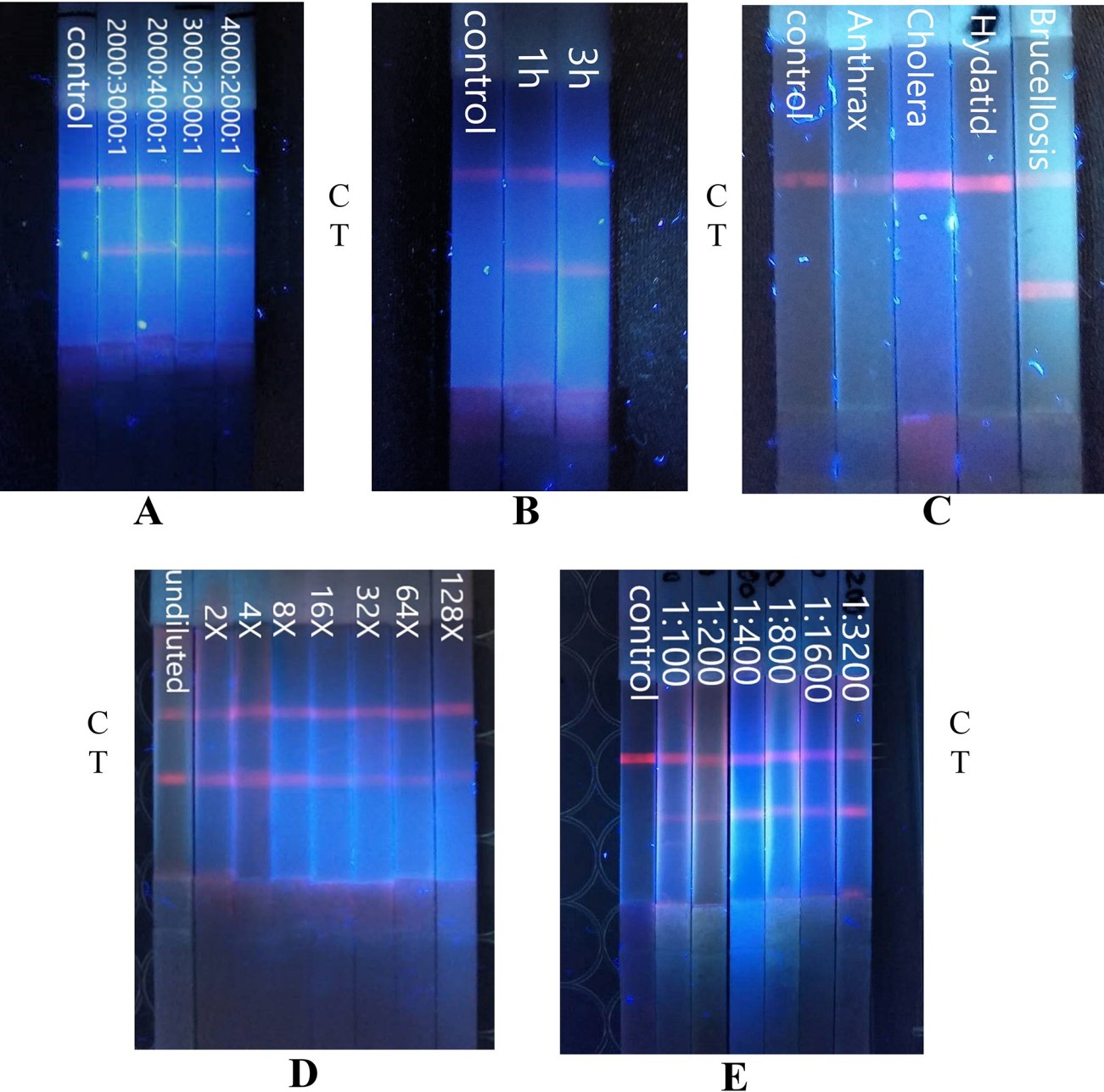

**Fig 2.** A) Effect of EDC:NHS:QD molar ratio on T-value. B) Effect of coupling duration on T-value. C) Specificity of QD test strips. D) Effect of dilution factor on T-value. E) Effect of serum titre on T-value.

amount of labelled *Brucella* total protein was tested at 10, 15 and 20 μg, and the T-line fluorescence value was strongest for 15 μg, followed by 20 μg and then 10 μg (Fig 3A).

## Selection and use of blocking agent

After QD-conjugated antigen was prepared, BSA was dissolved in glycine solution and casein solution as blocking agent to optimise the conditions. C- and T-lines appeared on the test strip under UV light with both solutions, which proved that the test strip was effective. The T-ray of

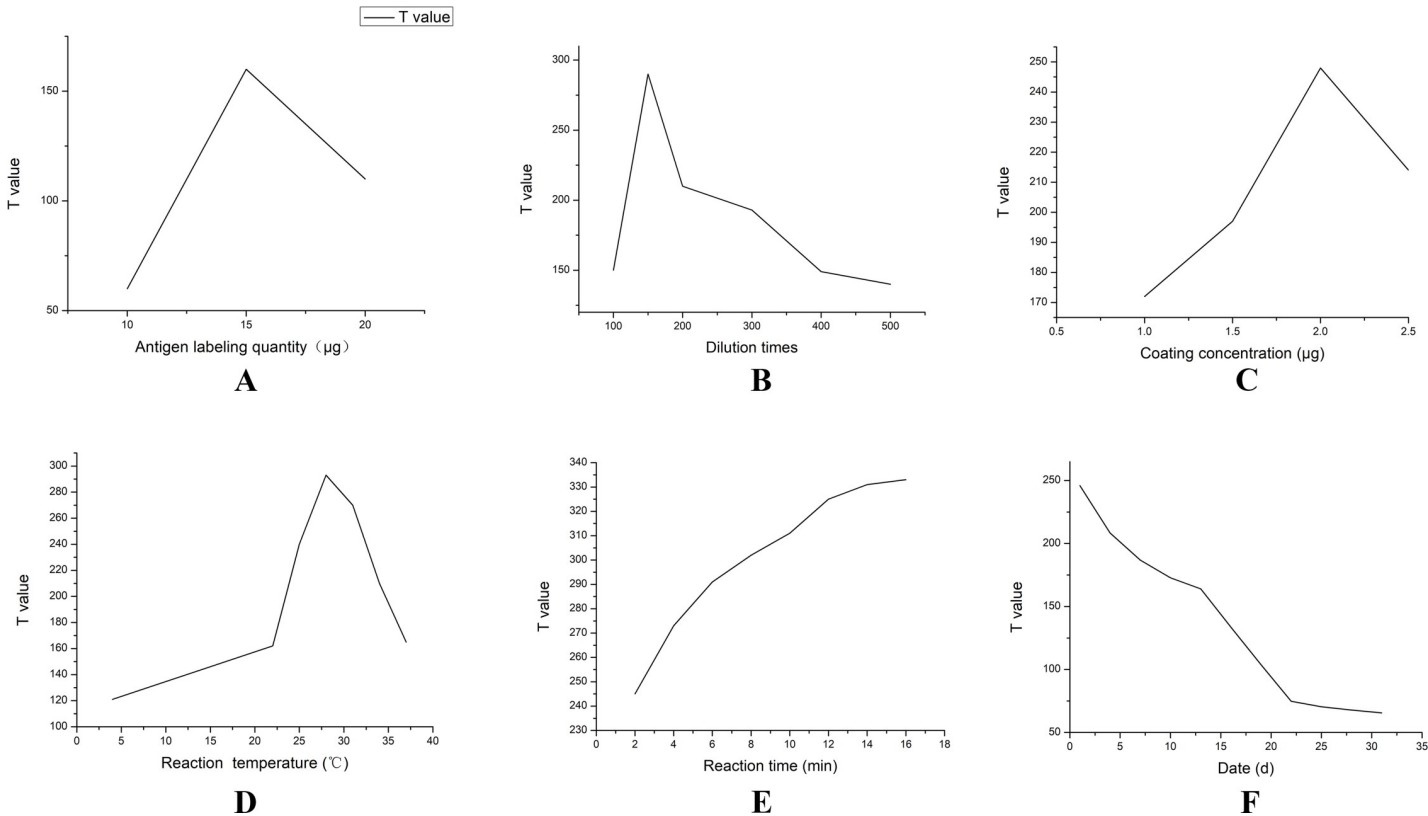

**Fig 3.** A) Effect of labelling amount on the T-value of test strips. B) Effect of conjugate pad dilution time on the T-value of test strips. C) Effect of different coating concentration on the T-value of test strips. D) Effect of reaction temperature on the T-line. E) Effect of reaction time on the T-line. F) Results of QD signal decay time.

the test strip using BSA solution was significantly brighter than that of the test strip using casein as a blocking agent. After screening the blocking agent type, the amount of blocking agent was optimised by testing 25 μl and 50 μl. When both C-lines appeared, the negative serum did not appear non-specifically. Comparing the test strips, it was obvious that the fluorescence intensity of with the larger volume of blocking agent (50 μl) was brighter than with 25 μl.

## Combined pad treatment and dilution testing

We observed the release of QD conjugates from the binding pad using two different drying methods, and the C-line appeared on both test strips. Release from the binding pad after drying was incomplete; a large number of QD conjugates remained attached, and the C-line was relatively faint. Freeze drying improved the release from the binding pad, since the brightness of the C-line was significantly greater than when using the other drying method. Thus, test strips treated with 5.0% sucrose and 0.6% Tween-20 yielded brighter fluorescence than the other two strips due to better release properties.

The fluorescence intensity of the T-value was strongest with a dilution factor of 150-fold (Fig 3B). As the dilution factor increased, the T-value gradually decreased.

## Optimisation of diluent and concentration for test strips

It is crucial to determine whether there is a blot on the NC membrane. The chromatographic membrane had obvious dark black stains when using the PBS dilution solution, while the

other two dilution solutions did not produce black stains. After addition of positive serum, the release speed of the test strips using methanol and PB was relatively fast, release from the binding pad was complete, and the chromatographic membrane did not retain excess conjugates, but this was not the case when using isopropyl alcohol and PBS as the dilution solution.

After negative serum was added dropwise, T-lines appeared under the UV lamp, indicating good specificity. After dripping the same type of positive serum, the C-line remained the same for all test strips, while the T-line changed colour. The T-value was most intense when the total bacterial protein coating concentration was 2.0 mg/mL, followed by 1.0 mg/mL and 2.5 mg/mL (Fig 3C).

## Assemble of test strip

The absorbent paper, NC membrane, bonding pad and sample pad were overlapped by 1 to 1.5 mm, pasted onto the PVC base plate, and pressed. The assembled test strip was then placed on an automatic cutting machine to generate strips 6.5 cm long and 3.5 mm wide. After assembly and cutting, strips were stored at room temperature for later use.

## Storage and reaction of test strips

Test strips were stored at 4, 27 and 37˚C. After a 10 min reaction, only the C-line was observed for negative controls at 4, 27 and 37˚C, indicating that the test strips achieved good specificity. Both positive and negative C-lines appeared, and the C-line of the test strip placed at 4˚C was darkest, followed by 37˚C and then 27˚C. The T-ray fluorescence intensity of the test strip placed at 27˚C was brighter than that at 4 and 37˚C.

Test strips were reacted at 4, 22, 25, 28, 31, 34 and 37˚C, and the results are shown in Fig 3D. When the reaction temperature was between 25 and 31˚C, the T-value of the test strip was large, and the fluorescence was bright. When the reaction temperature was lower than 25˚C or higher than 31˚C, the T-value decreased, and the fluorescence brightness gradually weakened.

After test strips were assembled, there was no non-specific appearance after addition of negative serum. After dropwise addition of positive serum, an immunofluorescence analyser was used to numerically assess test strips reacted for different reaction times every 2 min, during a total reaction time of 16 min. As can be seen from Fig 3E, the test paper changed over time; the value of the T-line gradually increased linearly over time. At ~10 min, the change in T-value was not obvious, and it essentially remained stable.

## Test strip specificity

Test strips were specifically verified with healthy human serum, anthrax serum, cholera serum and brucellosis-positive serum. After 10 min, a hand-held UV lamp was used to observe the signal on the test line. As shown in Fig 2C, after each type of serum was added, a C-line appeared on each test strip, which proves that each test strip was effective. There was no T-line with healthy human serum as well as anthrax and cholera serum samples, but there was a T-line on the test strip treated with brucellosis-positive serum.

Positive serum was diluted to a titre of 1:3200, and then diluted 2-, 4-, 8-, 16-, 32-, 64-, 128- and 256-fold. A hand-held UV lamp was then employed for qualitative observation. Stock serum (titre = 1:3200) and serum diluted 2 to 128 times yielded C- and T-lines. However, a T-line was not detected at a dilution of 256-fold, which indicates that the 128-fold dilution was the weakest signal that could be detected. Furthermore, when assessing the dilution, the detectable serum titre was 1:25 (Fig 2D).

Negative serum and different serum titres of 1:100, 1:200, 1:400, 1:800, 1:1600 and 1:3200 were added to the test strips, and observed with a handheld UV lamp. All strips yielded a C-

line, but negative serum did not produce a T-line while positive sera did yield a visible T-line at all titres tested. Furthermore, with increasing of serum titre, the fluorescence brightness of the T-line increased (Fig 2E).

The same positive serum was tested every 5 days, and the average value was calculated to be 274.8, with standard deviation of 11.29 and a coefficient of variation of 4.00%. Based on a coefficient of variation <15%, the test strip had a small coefficient of variation within 1 month, consistent with low variability and good stability.

Using the same batch of test strips, five negative sera and five standard positive sera were tested five times each. Negative sera only yielded a C-line, whereas positive sera produced both C and T lines, hence they were judged as positive. Different batches of test strips were tested against five negative and five standard positive sera. Negative sera only yielded a C-line, whereas positive sera produced both C- and T-lines, indicating that the test strip possessed good repeatability.

Five stored test strips were treated with positive serum, and numerical detection was performed every 2 days using an immunofluorescence analyser. It can be seen from Fig 3F that over time, the T-value of the test strip decreased, and the QD signal value was essentially stable up to ~25 days.

## Combination analysis

The 199 collected serum samples (135 positive and 64 negative by SAT) were tested by SAT and QD methods. The McNemar test results of the paired four-cell table data gave a P value of 0.688 ($P > 0.05$), a positive predictive value of 97.08% and a negative predictive value of 96.77%. The results of the QD test strips and SAT were not statistically significant (Table 1). Similarly, the diagnostic results of GICA and SAT, RBT and SAT, and QD test strip and GICA methods were not statistically significant.

The results of QD, GICA, RBT and SAT were analysed in series and in parallel to calculate the sensitivity, specificity, and Youden index (Table 2).

When RBT, GICA, QD, and SAT were analyzed in series, the sensitivity of QD and SAT in series was 98.52%, the specificity was 100%, and the Youden index was 0.99. We found that the specificity of QD and SAT when used in parallel was 93.75%, and the Youden index was 0.94. Thus, QD seems to have higher specificity when used in series with other tests.

## Discussion

The incidence of brucellosis is rising and this is a major public health concern. For the monitoring of this disease, SAT has disadvantages in terms of test duration, cost, sensitivity, the need for professional operators, and subjective assessment, all of which can impact the results. Colloidal gold immunochromatography has advantages both in terms of detection time and operation. However, evaluation of the quantum dot detection system in this research indicated that the signal value of the quantum dot is more stable. The T-line signal value of 25 days still

**Table 1. Results for QD test strip and SAT.**

| QD test strips | SAT | | |
|:---:|:---:|:---:|:---:|
| | **Positive** | **Negative** | **Total** |
| Positive | 133 | 4 | 137 |
| Negative | 2 | 60 | 62 |
| Total | 135 | 64 | 199 |

**Table 2. Screening results for combined tests.**

| Test | Serial test | | | Parallel test | | |
|------|-------------|---|---|---------------|---|---|
| | Se (%) | Sp (%) | Youden index | Se (%) | Sp (%) | Youden index |
| QD and SAT | 98.52 | 100 | 0.99 | 100 | 93.75 | 0.94 |
| GICA and SAT | 95.56 | 100 | 0.96 | 100 | 92.19 | 0.92 |
| RBT and SAT | 96.30 | 100 | 0.96 | 100 | 89.06 | 0.89 |
| QD and GICA | 94.16 | 100 | 0.94 | 100 | 91.94 | 0.92 |
| QD, GICA and SAT | 88.15 | 100 | 0.88 | 100 | 82.81 | 0.83 |
| GICA, RBT and SAT | 89.63 | 100 | 0.90 | 100 | 89.06 | 0.89 |

exists in our results. Herein, nanomaterial QDs were employed to develop a novel immuno-chromatography-based test that overcomes many of the deficiencies of the colloidal gold approach. The test facilitates simultaneous numerical detection, helping to avoid errors caused by human subjective judgments, and the detection results can be digitised to better support subsequent research. In this study, the signal value is basically detected by a fluorescent immunoassay analyzer.

Our test strip improves on existing QD labelling technology by optimizing the labelling conditions. Further, the use of QDs with a carboxylated CdSe/ZnS core shell structure with an excitation wavelength of 300−450 nm and an emission wavelength of 610 nm makes the results easy to visualize.

The stability of the test strip was investigated for 1 month, and all strips displayed good stability, with a coefficient of variation is less than 10%. Under normal circumstances, test strips need only be placed at normal temperature during the reaction process to achieve high fluorescence intensity, and test strips can also be stored at normal temperatures. In fact, temperatures above 37˚C may destroy total bacterial protein coated on the chromatography membrane, thereby inactivating the antigen, consequentially affecting the detection of serum. Repeatability testing of the same and different batches of test strips revealed good stability and repeatability.

Furthermore, the labelled antigen is *Brucella* total protein, the virulence of 104M is relatively weak, and it shares the same antigenic determinant as *B. melitensis*, so the risk of making antigens using the vaccine is low. This antigen is simply extracted from *Brucella* culture, has high antigen specificity, and can be used for differential diagnosis of other diseases with similar symptoms such as rheumatism at the epidemic site.

With regard to detection of *Brucella* antibodies by RBT, SAT and ELISA, the results showed that the sensitivity of ELISA and SAT was better than that of RBT in all three experiments; the coincidence rate of SAT and RBT detection methods was 96.3%. Based on China's Practice Guideline for brucellosis diagnosis and treatment, SAT was used as the gold standard for the diagnosis of brucellosis, In this study, analysis of the 135 positive and 64 negative samples using these three methods showed that more positive cases were detected by RBT and QD test strips than colloidal gold, and the number of test cases detected by QD test strips using negative serum was more than the other methods. The sensitivity and specificity of QD test strips were high, and the coincidence rate with SAT was also high, consistent with a high degree of coincidence, indicating their value for use in practice. In comparison with colloidal gold, quantum dots have a more stable signal value and a lower cost of preparation than colloidal gold.

In this study, quantum dot labelling technology and immunochromatography technology were successfully applied to create a quantum dot test strip that can qualitatively and numerically detect brucellosis antibodies in humans and animals, but a few limitations of this study

must be acknowledged. The first is that mainly human serum and sheep serum samples were used for the detection tests. Therefore, the efficiency of this test for other common hosts such as goats, cows, and swine has not been evaluated. The second limitation is that only 199 serum samples were tested. In the future, more samples across multiple centers need to be tested to determine the sensitivity and specificity in various contexts and patient types. Additionally, the test strip needs to be tested in epidemic regions, so as to assess its value as a practical initial screening test for brucellosis. Finally, an accelerated stability experiment also needs to be conducted with the test strip to determine the length of its shelf life.

In summary, this QD test strip does not require extensive laboratory infrastructure or technical expertise, and has high sensitivity and specificity, as well as a high coincidence rate with SAT.

## Acknowledgments

We are extremely grateful to the China Animal Disease Control Center and Academy of Military Medical Sciences.

## Author Contributions

**Conceptualization:** Guangqiang Li, Zhen Rong, Xiaowen Yang, Guozhong Tian, Hai Jiang.

**Data curation:** Guangqiang Li, Shengqi Wang, Dongri Piao, Hai Jiang.

**Investigation:** Zhen Rong, Shengqi Wang, Dongri Piao, Hai Jiang.

**Methodology:** Guangqiang Li, Hongyan Zhao, Hai Jiang.

**Project administration:** Hai Jiang.

**Resources:** Hongyan Zhao.

**Software:** Xiaowen Yang.

**Visualization:** Guozhong Tian.

**Writing – original draft:** Guangqiang Li, Hai Jiang.

**Writing – review & editing:** Guangqiang Li, Hai Jiang.

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
