## [Decision Letter · Decision Letter 0]

17 Apr 2020

Dear Mr. Jiang,

Thank you very much for submitting your manuscript "Rapid detection of brucellosis using a quantum dot-based immunochromatographic test strip" for consideration at PLOS Neglected Tropical Diseases. As with all papers reviewed by the journal, your manuscript was reviewed by members of the editorial board and by several independent reviewers. In light of the reviews (below this email), we would like to invite the resubmission of a significantly-revised version that takes into account the reviewers' comments. 

We cannot make any decision about publication until we have seen the revised manuscript and your response to the reviewers' comments. Your revised manuscript is also likely to be sent to reviewers for further evaluation.

Sincerely,

Joseph M. Vinetz

Deputy Editor

Joseph Vinetz

Deputy Editor

Reviewer's Responses to Questions

**Key Review Criteria Required for Acceptance?**

**Methods**

-Are the objectives of the study clearly articulated with a clear testable hypothesis stated?

-Is the study design appropriate to address the stated objectives?

-Is the population clearly described and appropriate for the hypothesis being tested?

-Is the sample size sufficient to ensure adequate power to address the hypothesis being tested?

-Were correct statistical analysis used to support conclusions?

-Are there concerns about ethical or regulatory requirements being met?

Reviewer #1: This is really a methods paper. Most of it is devoted to describing how they made the quantum dot assay. That could be of interest to some readers who may want to duplicate their test, but I think most would have wanted them to do more validation. The antigen is very poorly described; they say it is protein from Brucella 104M but they heated it at 180C, which likely denatured many proteins, and they don't provide a reference for the strain, nor a species name.

Reviewer #2: Objectives are good. Study design is appropriate. Source of sera need to be better defined. Sample size is adequate. Statistics appropriate. No ethical concerns. Suggestions:

 Introduction:

Lines 49-54: unnecessary

Line 60: two methods “culture and serology”

Line 69: Define “quantum dots”, then abbreviate (QD)

Line 71-74: clarify chemical elements of the periodic table Group IV….

Line 78: gold-labeled antibodies…

Line 80: clarify Stokes shift

Line 82: incomplete sentence

Line 86: …binds to a Brucella total protein preparation…

Materials and Methods:

Line 91: retrospective, “routinely collected” for what purposes?

Line 114: suggest omit “with good antigenicity”; also Brucella abortus vaccine strain 104M 

Line 115: grown on what medium?

Line 116: centrifugation (how fast, how long)?

Line 118: ..”more than twice… how determined?

Line 120: …”centrifuged, precipitated, and extracted” (specify)

Line 121: …”filtered”…(specify)

Line 140: sample of what?

Line 187: “brucellosis serum” better characterize

Reviewer #3: the study described the optimization process of newly designed diagnostic for human brucellosis and the authors claim a high diagnostic performance by comparison with several surogate reference test whereas at least an initial attempt should have undertaken to test culture confirmed samples (the accepted reference)

**Results**

-Does the analysis presented match the analysis plan?

-Are the results clearly and completely presented?

-Are the figures (Tables, Images) of sufficient quality for clarity?

Reviewer #1: The results include photos of the fluorescent test strips, but no information about the reader they used. They also do not say when in the course of the illness the + sera were collected; one can assume from the epidemiology of brucellosis in China that they describe in the introduction that they were all B. melitensis. Would infections with other species also be positive? B. canis infections are difficult to diagnose with standard serology.

Reviewer #2: Analysis needs work: Suggestions:

Results:

Line 228: I cannot find Fig. 5

Line 235: confusing

Line 252: NTC + NC ?

Line 298-299: better characterize the sera (assuming anthrax and cholera were post vaccination?)

Line 343: Fig. 1 difficult to interpret

Reviewer #3: I observed some discrepancies (a fig 5 mentioned is not presented) and a signal stated in fig 2d to be negative is clearly positive.

**Conclusions**

-Are the conclusions supported by the data presented?

-Are the limitations of analysis clearly described?

-Do the authors discuss how these data can be helpful to advance our understanding of the topic under study?

-Is public health relevance addressed?

Reviewer #1: partially. If the object was to make any easier to do or read serological assay they may have succeeded. They concluded that this is not a better test than currently available serologies.

Reviewer #2: Conclusions:

It might be of interest to compare this lateral flow assay with other lateral flow assays using other antigens. Also, since you are using brucella protein it would also be of interest to test against serum of patients with rough LPS such as B. canis.

Reviewer #3: (No Response)

**Editorial and Data Presentation Modifications?**

Reviewer #1: In Table 2 you need to define the abbreviations. For readers not familiar with QD you should explain why the preparation you made give better luminescence. You need to define your antigen better. How was protein eluted from the bacteria and polysaccharides excluded?

Reviewer #2: Manuscript can be shortened by 1/3 without changing results. In need of considerable editing.

Reviewer #3: The manuscript is rather lengthly and not very precise and may need rewriting by a more experienced scientist

**Summary and General Comments**

Reviewer #1: I believe they have technically improved the Quantum dot detection system but not made clear how this will improve the diagnosis o brucellosis.

Reviewer #2: Interesting application of new technology to Brucella diagnostics.

Reviewer #3: Testing of cuture confirmed samples is required

PLOS authors have the option to publish the peer review history of their article (what does this mean?). If published, this will include your full peer review and any attached files.

Reviewer #1: No

Reviewer #2: Yes: Edward J. Young, MD

Reviewer #3: No
---

## [Editor Report · Decision Letter 1]

2 Jul 2020

Dear Mr. Jiang,

We are pleased to inform you that your manuscript 'Rapid detection of brucellosis using a quantum dot-based immunochromatographic test strip' has been provisionally accepted for publication in PLOS Neglected Tropical Diseases.

Best regards,

Joseph M. Vinetz

Deputy Editor

Joseph Vinetz

Deputy Editor

---

## [Editor Report · Acceptance letter]

17 Sep 2020

Dear Mr. Jiang,

We are delighted to inform you that your manuscript, "Rapid detection of brucellosis using a quantum dot-based immunochromatographic test strip," has been formally accepted for publication in PLOS Neglected Tropical Diseases.

Best regards,

Shaden Kamhawi

co-Editor-in-Chief

Paul Brindley

co-Editor-in-Chief
